# Inequity aversion improves cooperation in intertemporal social dilemmas

**Edward Hughes,**[*] **Joel Z. Leibo,**[*] **Matthew Phillips, Karl Tuyls, Edgar Dueñez-Guzman,**
**Antonio García Castañeda, Iain Dunning, Tina Zhu, Kevin McKee, Raphael Koster,**
**Heather Roff, Thore Graepel**
DeepMind, London, United Kingdom
{edwardhughes, jzl, karltuyls, duenez, antoniogc, idunning, tinazhu,
kevinrmckee, rkoster, hroff, thore}@google.com,
matthew.phillips.12@ucl.ac.uk

## Abstract

Groups of humans are often able to find ways to cooperate with one another in complex, temporally extended social dilemmas. Models based on behavioral economics are only able to explain this phenomenon for unrealistic stateless matrix games. Recently, multi-agent reinforcement learning has been applied to generalize social dilemma problems to temporally and spatially extended Markov games. However, this has not yet generated an agent that learns to cooperate in social dilemmas as humans do. A key insight is that many, but not all, human individuals have inequity averse social preferences. This promotes a particular resolution of the matrix game social dilemma wherein inequity-averse individuals are personally pro-social and punish defectors. Here we extend this idea to Markov games and show that it promotes cooperation in several types of sequential social dilemma, via a profitable interaction with policy learnability. In particular, we find that inequity aversion improves temporal credit assignment for the important class of *intertemporal* social dilemmas. These results help explain how large-scale cooperation may emerge and persist.

## 1 Introduction

In intertemporal social dilemmas, there is a tradeoff between short-term individual incentives and long-term collective interest. Humans face such dilemmas when contributing to a collective food storage during the summer in preparation for a harsh winter, organizing annual maintenance of irrigation systems, or sustainably sharing a local fishery. Classical models of human behavior based on rational choice theory predict that cooperation in these situations is impossible [1, 2]. This poses a puzzle since humans evidently do find ways to cooperate in many everyday intertemporal social dilemmas, as documented by decades of fieldwork [3, 4] and laboratory experiments [5, 6]. Providing an empirically grounded explanation of how individual behavior gives rise to societal cooperation is seen as a core goal in several subfields of the social sciences and evolutionary biology [7, 8, 9].

[10, 11] proposed influential models based on behavioral game theory. However, these models have limited applicability since they only generate predictions when the problem can be cast as a matrix game (see e.g. [12, 13]). Here we consider a more realistic video-game setting, like those introduced in the behavioral research of [14, 15, 16]. In this environment, agents do not simply choose to cooperate or defect like they do in matrix games. Rather they must learn policies to implement their strategic decisions, and must do so while coping with the non-stationarity arising from other agents learning simultaneously. Several papers used multi-agent reinforcement learning [17, 18, 19] and

---

[*]Equal contribution.

planning [20, 21, 22, 23] to generate cooperation in this setting. However, this approach has not yet demonstrated robust cooperation in games with more than two players, which is often observed in human behavioral experiments. Moreover naïvely optimizing group reward is also ineffective, due to the lazy agent problem [24].[†]

It is difficult for both natural and artificial agents to find cooperative solutions to intertemporal social dilemmas for the following reasons:

1. Collective action – individuals must learn and coordinate policies at a group level to avoid falling into socially deficient equilibria.
2. Temporal credit assignment – rational defection in the short-term must become associated with long-term negative consequences.

Many different research traditions, including economics, evolutionary biology, sociology, psychology, and political philosophy have all converged on the idea that fairness norms are involved in resolving social dilemmas [25, 26, 27, 28, 29, 30, 31]. In one well-known model, agents are assumed to have inequity-averse preferences [10]. They balance their selfish desire for individual rewards against a need to keep deviations between their own rewards and the rewards of others as small as possible. Inequity-averse individuals are able to solve social dilemmas by resisting the temptation to pull ahead of others or—if punishment is possible—by punishing and discouraging free-riding. The inequity aversion model has been successfully applied to explain human behavior in a variety of laboratory economic games, such as the ultimatum game, the dictator game, the gift exchange game, market games, the trust game and public goods [32, 33].[‡]

In this research, we generalize the inequity aversion model to Markov games, and show that it resolves intertemporal social dilemmas. Crucial to our analysis will be the distinction between *disadvantageous* inequity aversion (negative reward received by individuals who underperform relative to others) and *advantageous* inequity aversion (negative reward received by individuals who overperform relative to others). Colloquially, these may be thought of as reductionist models of envy (disadvantageous inequity aversion) and guilt (advantageous inequity aversion) respectively [36]. We hypothesise that these directly address the two challenges set out above in the following way.

Inequity aversion mitigates the problem of collective action by changing the effective payoff structure experienced by agents through both a direct and an indirect mechanism. In the direct mechanism, defectors experience advantageous inequity aversion, diminishing the marginal benefit of defection over cooperation. The indirect mechanism arises when cooperating agents are disadvantageous-inequity averse. This motivates them to punish defectors by sanctioning them, reducing the payoff incentive for free-riding. Since agents must learn a defecting strategy via exploration, initially cooperative agents are deterred from switching strategies if the payoff bonus does not outweigh the cost of inefficiently executing the defecting strategy while learning.

Inequity aversion also ameliorates the temporal credit assignment problem. Learning the association between short-term actions and long-term consequences is a high-variance and error-prone process, both for animals [37] and reinforcement learning algorithms [38]. Inequity aversion short-circuits the need for such long-term temporal credit assignment by acting as an "early warning system" for intertemporal social dilemmas. As before, both a direct and an indirect mechanism are at work. With the direct mechanism, advantageous-inequity-averse defectors receive negative rewards in the short-term, since the benefits of defection are delivered on that timescale. The indirect mechanism operates because cooperators experience disadvantageous inequity aversion at precisely the time when other agents defect. This leads cooperators to punish defectors on a short-term timescale. Both systems have the effect of operant conditioning [39], incentivizing agents that cannot resolve long-term uncertainty to act in the lasting interest of the group.

## 2 Reinforcement learning in sequential social dilemmas

### 2.1 Partially observable Markov games

We consider multi-agent reinforcement learning in partially-observable general-sum Markov games [40, 41]. In each game state, agents take actions based on a partial observation of the state space and

---

[†]For more detail on the motivations for our research program, see the supplementary information.
[‡]For alternative theories of the other-regarding preferences that may underlie human cooperative behavior in economic games, see [34, 35].

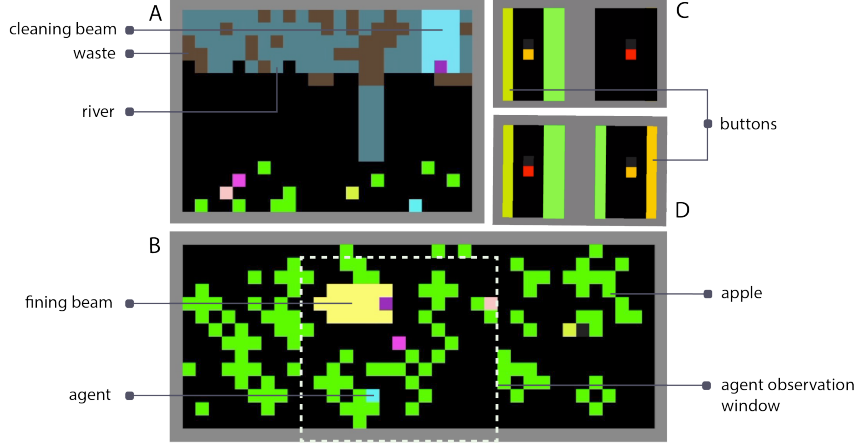

Figure 1: Screenshots from (A) the Cleanup game, (B) the Harvest game, (C) the Dictate apples game, and (D) the Take apples and Give apples games. The size of the agent-centered observation window is also shown in (B). The same size observation was used in all experiments.

receive an individual reward. Agents must learn through experience an appropriate behavior policy while interacting with one another. We formalize this as follows.

Consider an $N$-player partially observable Markov game $\mathcal{M}$ defined on a finite set of states $\mathcal{S}$. The observation function $O : \mathcal{S} \times \{1, \ldots, N\} \to \mathbb{R}^d$ specifies each player's $d$-dimensional view on the state space. From each state, players may take actions from the set $\mathcal{A}^1, \ldots, \mathcal{A}^N$ (one for each player). As a result of their joint action $a^1, \ldots, a^N \in \mathcal{A}^1, \ldots, \mathcal{A}^N$ the state changes following the stochastic transition function $\mathcal{T} : \mathcal{S} \times \mathcal{A}^1 \times \cdots \times \mathcal{A}^N \to \Delta(\mathcal{S})$ (where $\Delta(\mathcal{S})$ denotes the set of discrete probability distributions over $\mathcal{S}$). Write $\mathcal{O}^i = \{o^i \mid s \in \mathcal{S}, o^i = O(s, i)\}$ to indicate the observation space of player $i$. Each player receives an individual extrinsic reward defined as $r^i : \mathcal{S} \times \mathcal{A}^1 \times \cdots \times \mathcal{A}^N \to \mathbb{R}$ for player $i$.[§]

Each agent learns, independently through its own experience of the environment, a behavior policy $\pi^i : \mathcal{O}^i \to \Delta(\mathcal{A}^i)$ (written $\pi(a^i|o^i)$) based on its own observation $o^i = O(s, i)$ and extrinsic reward $r^i(s, a^1, \ldots, a^N)$. For the sake of simplicity we will write $\vec{a} = (a^1, \ldots, a^N)$, $\vec{o} = (o^1, \ldots, o^N)$ and $\vec{\pi}(.|\vec{o}) = (\pi^1(.|o^1), \ldots, \pi^N(.|o^N))$. Each agent's goal is to maximize a long term $\gamma$-discounted payoff defined as follows:

$$V^i_{\vec{\pi}}(s_0) = \mathbb{E}\left[\sum_{t=0}^{\infty} \gamma^t r^i(s_t, \vec{a}_t) | \vec{a}_t \sim \vec{\pi}_t, s_{t+1} \sim \mathcal{T}(s_t, \vec{a}_t)\right] . \tag{1}$$

## 2.2 Learning agents

We deploy asynchronous advantage actor-critic (A3C) as the learning algorithm for our agents [42]. A3C maintains both value (critic) and policy (actor) estimates using a deep neural network. The policy is updated according to the policy gradient method, using a value estimate as a baseline to reduce variance. Gradients are generated asynchronously by 24 independent copies of each agent, playing simultaneously in distinct instantiations of the environment. Explicitly, the gradients are $\nabla_\theta \log \pi(a_t|s_t; \theta) A(s_t, a_t; \theta, \theta_v)$, where $A(s_t, a_t; \theta, \theta_v)$ is the advantage function, estimated via $k$-step backups, $\sum_{i=0}^{k-1} \gamma^i u_{t+i} + \gamma^k V(s_{t+k}; \theta_v) - V(s_t; \theta_v)$ where $u_{t+i}$ is the *subjective* reward. In section 3.1 we decompose this into an *extrinsic* reward from the environment and an *intrinsic* reward that defines the agent's inequity-aversion.

## 2.3 Intertemporal social dilemmas

An intertemporal social dilemma is a temporally extended multi-agent game in which individual short-term optimal strategies lead to poor long-term outcomes for the group. To define this term

---

[§]In our games, $N = 5$, $d = 15 \times 15 \times 3$ and $|\mathcal{A}^i|$ ranges from 8 to 10, with actions comprising movement, rotation and firing.

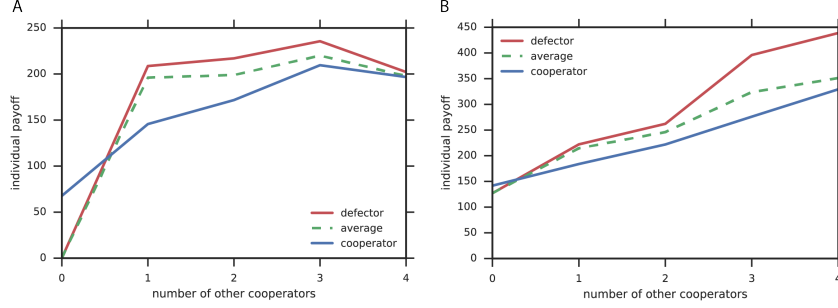

Figure 2: The public goods game (Cleanup) and the commons game (Harvest) are social dilemmas. (A) shows the Schelling diagram for Cleanup. (B) shows the Schelling diagram for Harvest. The dotted line shows the overall average return were the individual to choose defection.

precisely, we employ a formalization of empirical game theoretic analysis [43, 44]. Our definition is consistent with that of [17]. However, since that work was limited to the 2-player case, it relied on the empirical payoff matrix to represent the relative values of cooperation and defection. This quantity is unwieldy for $N > 2$ since it becomes a tensor. Therefore we base our definition on a different representation of the $N$-player game. Explicitly, a *Schelling diagram* [45, 18] depicts the relative payoffs for a single cooperator or defector given a fixed number of other cooperators. Thus Schelling diagrams are a natural and convenient generalization of payoff matrices to multi-agent settings. Game-theoretic properties like Nash equilibria are readily visible in Schelling diagrams; see [45] for additional details and intuition.

An $N$-player *sequential social dilemma* is a tuple $(\mathcal{M}, \Pi = \Pi_c \sqcup \Pi_d)$ of a Markov game and two disjoint sets of policies, said to implement cooperation and defection respectively, satisfying the following properties. Consider the strategy profile $(\pi_c^1, \ldots, \pi_c^\ell, \pi_d^1, \ldots, \pi_d^m) \in \Pi_c^\ell \times \Pi_d^m$ with $\ell + m = N$. We shall denote the average payoff for the cooperating policies by $R_c(\ell)$ and for the defecting policies by $R_d(\ell)$. A *Schelling diagram* plots the curves $R_c(\ell + 1)$ and $R_d(\ell)$. Intuitively, the diagram displays the two possible payoffs to the $N^{\text{th}}$ player given that $\ell$ of the remaining players elect to cooperate and the rest defect. We say that $(\mathcal{M}, \Pi)$ is a sequential social dilemma iff the following hold:

1. Mutual cooperation is preferred over mutual defection: $R_c(N) > R_d(0)$.
2. Mutual cooperation is preferred to being exploited by defectors: $R_c(N) > R_c(0)$.
3. Either the *fear* property, the *greed* property, or both:
   - Fear: mutual defection is preferred to being exploited. $R_d(i) > R_c(i)$ for sufficiently small $i$.
   - Greed: exploiting a cooperator is preferred to mutual cooperation. $R_d(i) > R_c(i)$ for sufficiently large $i$.

We show that the matrix games Stag Hunt, Chicken and Prisoner's Dilemma satisfy these properties in Supplementary Fig. 1.

A sequential social dilemma is *intertemporal* if the choice to defect is optimal in the short-term. More precisely, consider an individual $i$ and an arbitrary set of policies for the rest of the group. Given a starting state, for all $k$ sufficiently small, the policy $\pi_k^i \in \Pi$ with maximum return in the next $k$ steps is a defecting policy. There is thus a tension between short-term personal gain and long-term group utility.

## 2.4 Examples

[46] divides all multi-person social dilemmas into two broad categories:

1. *Public goods dilemmas*, in which an individual must pay a personal cost in order to provide a resource that is shared by all.
2. *Commons dilemmas*, in which an individual is tempted by a personal benefit, depleting a resource that is shared by all.

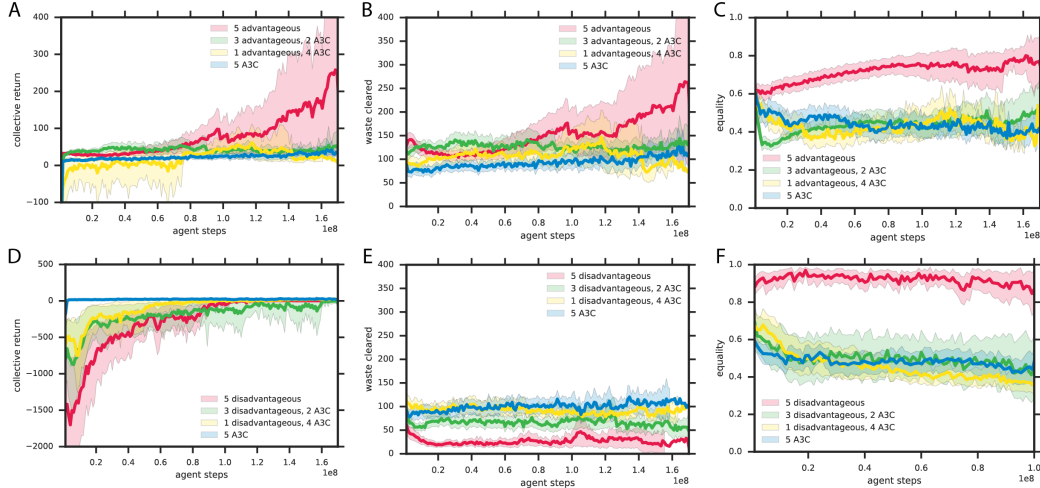

Figure 3: Advantageous inequity aversion facilitates cooperation in the Cleanup game. (A) compares the collective return achieved by A3C and advantageous inequity averse agents, (B) shows contributions to the public good, and (C) shows equality over the course of training. (D-F) demonstrate that disadvantageous inequity aversion does not promote greater cooperation in the Cleanup game.

We consider two dilemmas in this paper, one of the public goods type and one of the commons type. Each was implemented as a partially observable Markov game on a 2D grid. Both are also intertemporal social dilemmas because individually selfish actions produce immediate benefits while their impacts on the collective develop over a longer time horizon. The availability of costly punishment is of critical importance in human sequential social dilemmas [47, 48] and is therefore an action in the environments presented here.¶

In the *Cleanup* game, the aim is to collect apples from a field. Each apple provides a reward of 1. The spawning of apples is controlled by a geographically separate aquifer that supplies water and nutrients. Over time, this aquifer fills up with waste, lowering the respawn rate of apples linearly. For sufficiently high waste levels, no apples can spawn. At the start of each episode, the environment resets with waste just beyond this saturation point. To cause apples to spawn, agents must clean some of the waste.

Here we have a dilemma. Provided that some agents contribute to the public good by cleaning up the aquifer, it is individually more rewarding to stay in the apple field. However, if all players defect, then no-one gets any reward. A successful group must balance the temptation to free-ride with the provision of the public good. Cooperative agents must make a positive commitment to group-level well-being to solve the task.

The goal of the *Harvest* game is to collect apples. Each apple provides a reward of 1. The apple regrowth rate varies across the map, dependent on the spatial configuration of uncollected apples: the more nearby apples, the higher the local regrowth rate. If all apples in a local area are harvested then none ever grow back. After 1000 steps the episode ends, at which point the game resets to an initial state.

The dilemma is as follows. The short-term interests of each individual leads toward harvesting as rapidly as possible. However, the long-term interests of the group as a whole are advanced if individuals refrain from doing so, especially when many agents are in the same local region. Such situations are precarious because the more harvesting agents there are, the greater the chance of permanently depleting the local resources. Cooperators must abstain from a personal benefit for the good of the group.‖

---

¶In both games, players can fine each other using a punishment beam. This contrasts with [18], in which a timeout beam was used.

‖Precise details of the ecological dynamics may be found in the supplementary information.

## 2.5 Validating the environments

We would like to demonstrate that these environments are social dilemmas by plotting Schelling diagrams. In complex, spatially and temporally extended Markov games, it is not feasible to analytically determine cooperating and defecting policies. Instead, we must study the environment empirically. One method employs reinforcement learning to train such policies. We enforce cooperation or defection by making appropriate modifications to the environment, as follows.

In Harvest, we enforce cooperation by modifying the environment to prevent some agents from gathering apples in low-density areas. In Cleanup, we enforce free-riding by removing the ability of some agents to clean up waste. We also add a small group reward signal to encourage the remaining agents to cooperate. The resulting empirical Schelling diagrams in Figure 2 prove that our environments are indeed social dilemmas.

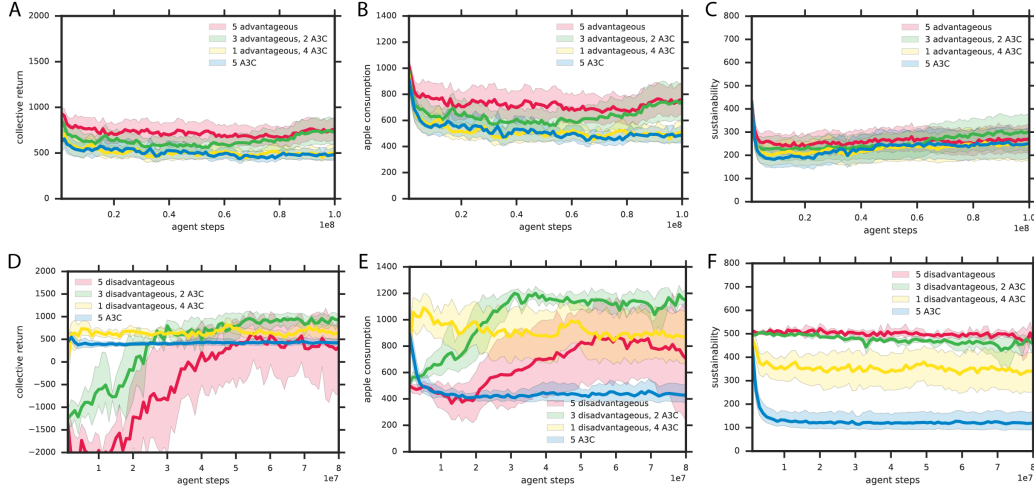

Figure 4: Inequity aversion promotes cooperation in the Harvest game. When all 5 agents have advantageous inequity aversion, there is a small improvement over A3C in the three social outcome metrics: (A) collective return, (B) apple consumption, and (C) sustainability. Disadvantageous inequity aversion provides a much larger improvement over A3C, and works even when only 1 out of 5 agents are inequity averse. (D) shows collective return, (E) apple consumption, and (F) sustainability.

## 3 The model

We first introduce the inequity aversion model of [10]. It is directly applicable only to stateless games. We then extend their model to sequential or multi-state problems, making use of deep reinforcement learning.

### 3.1 Inequity aversion

The [10] utility function is as follows. Let $r_1, \ldots, r_N$ be the extrinsic payoffs achieved by each of $N$ players. Each agent receives a utility

$$U_i(r_i, \ldots r_N) = r_i - \frac{\alpha_i}{N-1} \sum_{j \neq i} \max\left(r_j - r_i, 0\right) - \frac{\beta_i}{N-1} \sum_{j \neq i} \max\left(r_i - r_j, 0\right), \quad (2)$$

where the additional terms may be interpreted as intrinsic payoffs, in the language of [49].

The parameter $\alpha_i$ controls an agent's aversion to *disadvantageous* inequity. A larger value for $\alpha_i$ implies a larger utility loss when other agents achieve rewards greater than one's own. Likewise, the parameter $\beta_i$ controls an agent's aversion to *advantageous* inequity, utility lost when performing better than others. [10] argue that $\alpha > \beta$. That is, most people are loss averse in social comparisons. There is some empirical support for this prediction [50], though the evidence is mixed [51, 52]. In a sweep over values for $\alpha$ and $\beta$, we found our strongest results for $\alpha = 5$ and $\beta = 0.05$.

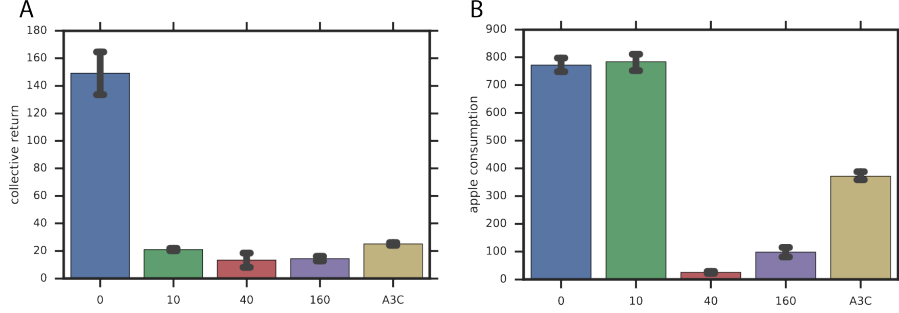

Figure 5: Inequity aversion promotes cooperation by improving temporal credit assignment. (A) shows collective return for delayed advantageous inequity aversion in the Cleanup game. (B) shows apple consumption for delayed disadvantageous inequity aversion in the Harvest game.

## 3.2 Inequity aversion in sequential dilemmas

Experimental work in behavioral economics suggests that some proportion of natural human populations are inequity averse [8]. However, as a computational model, inequity aversion has only been expounded for the matrix game setting. Equation (2) can be directly applied only to stateless games [53, 54]. In this section we extend this model of inequity aversion to the temporally extended Markov game case.

The main problem in re-defining the social preference of equation (2) for Markov games is that the rewards of different players may occur on different timesteps. Thus the key step in extending (2) to this case is to introduce per-player temporal smoothing of the reward traces.

Let $r_i(s, a)$ denote the reward obtained by the $i$-th player when it takes action $a$ from state $s$. For convenience, we also sometimes write it with a time index: $r_i^t := r_i(s^t, a^t)$. We define the subjective reward $u_i(s, a)$ received by the $i$-th player when it takes action $a$ from state $s$ to be

$$u_i(s_i^t, a_i^t) = r_i(s_i^t, a_i^t) - \frac{\alpha_i}{N-1} \sum_{j \neq i} \max(e_j^t(s_j^t, a_j^t) - e_i^t(s_i^t, a_i^t), 0) \qquad (3)$$
$$- \frac{\beta_i}{N-1} \sum_{j \neq i} \max(e_i^t(s_i^t, a_i^t) - e_j^t(s_j^t, a_j^t), 0),$$

where the temporal smoothed rewards $e_j^t$ for the agents $j = 1, \ldots, N$ are updated at each timestep $t$ according to

$$e_j^t(s_j^t, a_j^t) = \gamma \lambda e_j^{t-1}(s_j^{t-1}, a_j^{t-1}) + r_j^t(s_j^t, a_j^t), \qquad (4)$$

where $\gamma$ is the discount factor and $\lambda$ is a hyperparameter. This is analogous to the mathematical formalism used for eligibility traces [55]. Furthermore, we allow agents to observe the smoothed reward of every player on each timestep.

## 4 Results

We show that advantageous inequity aversion is able to resolve certain intertemporal social dilemmas without resorting to punishment by providing a temporally correct intrinsic reward. For this mechanism to be effective, the population must have sufficiently many advantageous-inequity-averse individuals. By contrast disadvantageous-inequity-averse agents can drive mutual cooperation even in small numbers. They achieve this by punishing defectors at a time concomitant with their offences. In addition, we find that advantageous inequity aversion is particularly effective for resolving public goods dilemmas, whereas disadvantageous inequity aversion is more powerful for addressing commons dilemmas. Our baseline A3C agent fails to find socially beneficial outcomes in either category of game. We define the metrics used to quantify our results in the supplementary information.

### 4.1 Advantageous inequity aversion promotes cooperation

Advantageous-inequity-averse agents are better than A3C at maintaining cooperation in both public goods and commons games. This effect is particularly pronounced in the Cleanup game (Figure 3).

Here groups of 5 advantageous-inequity-averse agents find solutions in which 2 consistently clean large amounts of waste, producing a large collective return.** We clarify the effect of advantageous inequity aversion on the intertemporal nature of the problem by delaying the delivery of the intrinsic reward signal. Figure 5 suggests that improving temporal credit assignment is an important function of inequity aversion since delaying the time at which the intrinsic reward signal is delivered removes its beneficial effect.

### 4.2 Disadvantageous inequity aversion promotes cooperation

Disadvantageous-inequity-averse agents are better than A3C at maintaining cooperation via punishment in commons games (Figure 4). In particular, a single disadvantageous-averse agent can fine defectors, generating a sustainable outcome.†† In Figure 5, we see that the disadvantageous-inequity-aversion signal must be temporally aligned with over-consumption for effective policing to arise. Hence, it is plausible that inequity aversion bridges the temporal gap between short-term incentives and long-term outcomes. Disadvantageous inequity aversion has no such positive impact in the Cleanup game, for reasons that we discuss in section 5.

## 5 Discussion

In the Cleanup game, advantageous inequity aversion is an unambiguous feedback signal: it encourages agents to contribute to the public good. In the direct pathway, trial and error will quickly discover that the fastest way to diminish the negative rewards arising from advantageous inequity aversion is to clean up waste, since doing so creates more apples for others to consume. However the indirect mechanism of disadvantageous inequity aversion and punishment lacks this property; while punishment may help exploration of new policies, it does not directly increase the attractiveness of waste cleaning.

The Harvest game requires passive abstention rather than active provision. In this setting, advantageous inequity aversion provides a noisy signal for sustainable behaviour. This is because it is sensitive to the precise apple configuration in the environment, which changes rapidly over time. Hence advantageous inequity aversion does not greatly aid the exploration of policy space. Punishment, on the other hand, operates as a valuable shaping reward for learning, dis-incentivizing overconsumption at precisely the correct time and place.

In the Harvest game, disadvantageous inequity aversion generates cooperation in a grossly inefficient manner: huge amounts of collective resource are lost to fines (compare Figures 4D and 4E). This parallels human behavior in laboratory matrix games, e.g. [56, 57]. In the Cleanup game, advantageous-inequity averse agents resolve the social dilemma without such losses, but must comprise a large proportion of the population to be successful. This mirrors the cultural modulation of advantageous inequity aversion in humans [58]. Evolution is hypothesized to have favored fairness as a mechanism for continued human cooperation [59]. It remains to be seen whether emergent inequity-aversion can be obtained by evolving reinforcement learning agents.

We conclude by putting our approach in the context of prior work. Since our mechanism does not require explicitly training cooperating and defecting agents or modelling their behaviour, it scales more easily to complex environments and large populations of agents. However, our method has several limitations. Firstly, our guilty agents are quite exploitable, as evidenced by the necessity of a homogeneous guilty population to achieve cooperation. Secondly, our agents use outcomes rather than predictions to inform their policies. This is known to be a problem in environments with high stochasticity [22]. Finally, the heterogeneity of the population is an additional hyperparameter in our model. Clearly, one must set this appropriately, particularly in games with asymmetric outcomes. It is likely that a hybrid approach will be required to solve these challenging issues at scale.

---

**For a video of this behavior, visit `https://youtu.be/N8BUzzFx7uQ`.

††For a video of this behavior, visit `https://youtu.be/tz3ZpTTmxTk`.

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
