[Supplementary Material · Inequity aversion improves cooperation in intertemporal social dilemmas (Supplementary).pdf]

# Inequity aversion improves cooperation in intertemporal social dilemmas

**Edward Hughes,**[*] **Joel Z. Leibo,**[*] **Matthew Phillips, Karl Tuyls, Edgar Dueñez-Guzman,**
**Antonio García Castañeda, Iain Dunning, Tina Zhu, Kevin McKee, Raphael Koster,**
**Heather Roff, Thore Graepel**
DeepMind, London, United Kingdom
{edwardhughes, jzl, karltuyls, duenez, antoniogc, idunning, tinazhu,
kevinrmckee, rkoster, hroff, thore}@google.com,
matthew.phillips.12@ucl.ac.uk

## A  Supplementary information

### A.1  Motivating research on emergent cooperation

The aims of this new research program are twofold. First, we seek to better understand the individual level inductive biases that promote emergent cooperation at the group level in humans. Second, we want to develop agents that exhibit these inductive biases, in the hope that they might navigate complex multi-agent tasks in a human-like way. Much as the fields of neuroscience and reinforcement learning have enjoyed a symbiotic relationship over the past fifty years, so also can behavioral economics and multi-agent reinforcement learning.

Consider, for comparison, maximizing joint utility. Firstly, this assumes away the problem of emergent altruism on the individual level, which is exactly our object of study. Therefore, it is not a relevant baseline for our research. Moreover, it is known to suffer from a serious spurious reward problem (Sunehag et al. 2017), which gets worse as the number of agents increases. Furthermore, in realistic environments, one may not have access to the collective reward function, for privacy reasons for example. Finally, groups of agents trained with a group reward are by definition overfitting to the outcomes of their co-players. Thus maximizing joint utility does not easily generalize to complicated multi-agent problems with large numbers of agents and subtasks that mix cooperation and competition.

Individual-level inductive biases sidestep these issues, while allowing us to learn from the extensive human behavioral literature. In this paper, we have taken an extremely well-studied model in the game-theoretic setting (Fehr and Schmidt 1999) and recast it as an intrinsic reward for reinforcement learning. We can thus evaluate the strengths and weaknesses of inequity aversion from a completely new perspective. We note its success in solving social dilemmas, but find that the success is task-conditional, and that the policies are sometimes quite exploitable. This suggests various fascinating extensions, such as a population-based study with evolved intrinsic rewards (Wang et al. to appear).

### A.2  Illustrative Schelling diagrams for 2-player matrix games and SSDs

Figure 1 shows Schelling diagrams and the associated payoff matrices for the canonical matrix games Chicken, Stag Hunt and Prisoner's Dilemma. We may read off the pure strategy Nash equilibria by considering the social pressure generated by the dominant strategy. Where this is defection, then there is a negative pressure on the number of cooperators; where this is cooperation, there is a positive pressure. Hence the pure strategy Nash equilibria in Chicken are $(c, d)$ and $(d, c)$, in Stag Hunt $(c, c)$ and $(d, d)$ and in Prisoner's Dilemma $(d, d)$. Moreover, the different motivations for defection are

---

[*]Equal contribution.

immediately apparent. In Chicken, greed promotes defection: $R_d(1) > R_c(1)$. In Stag Hunt, the problem is fear: $R_d(0) > R_c(0)$. Prisoner's Dilemma suffers from both temptations to defect.

## A.3   Parameters for Cleanup and Harvest games

In both Cleanup and Harvest, all agents are equipped with a fining beam which administers $-1$ reward to the user and $-50$ reward to the individual that is being fined. There is no penalty to the user for unsuccessful fining. In Cleanup each agent is additionally equipped with a cleaning beam, which allows them to remove waste from the aquifer. In both games, eating apples provides a reward of $1$. There are no other extrinsic rewards.

In Cleanup, waste is produced uniformly in the river with probability $0.5$ on each timestep, until the river is saturated with waste, which happens when the waste covers 40% of the river. For a given saturation $x$ of the river, apples spawn in the field with probability $0.125x$. Initially the river is saturated with waste, so some contribution to the public good is required for any agent to receive a reward.

Figure 1: These Schelling diagrams demonstrate that classic matrix games are social dilemmas by our definition.

In Harvest, apples spawn relative to the current number of other apples within an $\ell^1$ radius of 2. The spawn probabilities are $0, 0.005, 0.02, 0.05$ for $0, 1, 2$ and $\geq 3$ apples inside the radius respectively. The initial distribution of apples creates a number of more or less precariously linked regions. Sustainable policies must preferentially harvest denser regions, and avoid removing the important apples that link patches.

## A.4   Social outcome metrics

Unlike in single-agent reinforcement learning where the value function is the canonical metric of agent performance, in multi-agent systems with mixed incentives, there is no scalar metric that can adequately track the state of the system (see e.g. [1, 2]). Thus we use several different social outcome metrics in order to summarize group behavior and facilitate its analysis.

Consider $N$ independent agents. Let $\{r_t^i \mid t = 1, \ldots, T\}$ be the sequence of rewards obtained by the $i$-th agent over an episode of duration $T$. Likewise, let $\{o_t^i \mid t = 1, \ldots T\}$ be the $i$-th agent's observation sequence. Its return is given by $R^i = \sum_{t=1}^{T} r_t^i$.

The *Utilitarian metric* ($U$), also known as *collective return*, measures the sum total of all rewards obtained by all agents. It is defined as the average over players of sum of rewards $R^i$. The *Equality* metric ($E$) is defined using the Gini coefficient [3]. The *Sustainability* metric ($S$) is defined as the average time at which the rewards are collected. For the Cleanup game, we also consider a measure of total contribution to the public good ($P$), defined as the number of waste cells cleaned.

$$U = \mathbb{E}\left[\frac{\sum_{i=1}^{N} R^i}{T}\right], \tag{1}$$

$$E = 1 - \frac{\sum_{i=1}^{N} \sum_{j=1}^{N} |R^i - R^j|}{2N \sum_{i=1}^{N} R^i}, \tag{2}$$

$$S = \mathbb{E}\left[\frac{1}{N} \sum_{i=1}^{N} t^i\right] \quad \text{where} \ \ t^i = \mathbb{E}[t \mid r_t^i > 0], \tag{3}$$

$$P = \sum_{i}^{N} p_i . \tag{4}$$

where $p_i$ is the number of waste cells cleaned by player $i$.

## A.5 Dictate apples, Give apples and Take apples games

In each game, two players are isolated from one another in separate "rooms". They can interact only by pressing buttons. In the *Dictate* apples game, initially all apples are in the left room. At any time, the left agent can press a button that transports all the apples it has not yet consumed to the right room. In the *Take* apples game, both players begin with apples in their room, but there are twice as many in the left room as the right room. The right agent has the option at any time of pressing a button that removes all the apples from the other player's room that have not yet been collected. In the *Give* apples game, both players begin with apples, and the left player again has twice as many as the right player. The left player can press a button to add more apples on the right side. Unlike in the Dictate apples game, this has no effect on the left agent's own apple supply. Each episode terminates when all apples are collected.

## A.6 Inequity aversion models "irrational" behavior

The inequity aversion model of [4] is supported by experimental evidence from behavioral game theory. In particular, human behavior in the Dictator game is consistent with the prediction that some people have inequity-averse social preferences. A subject in a typical Dictator game experiment must decide how much of an initial endowment (if any) to give to another subject in a one-shot anonymous manner. In contrast to the prediction of rational choice theory that subjects would offer 0—but in accord with the prediction of [4]'s inequity aversion model—most subjects offer between $10\%$ and $50\%$ [5].

Figure 2: Behavioral economics laboratory paradigms can be simulated by gridworld Markov games. Agent behavior is shown in (A) for the Dictate apples game, in (B) for the Take apples game, and in (C) for the Give apples game.

To test whether our temporally extended inequity-aversion model makes predictions consistent with these findings, we introduce 3 simple 2-player gridworld games (see Figure **??**). These capture the essential features of Dictator game laboratory experiments. As in all our experiments, positive agent external rewards can only be obtained by collecting apples. In addition an agent can press buttons which *Dictate* apples (give from its own store), *Give* apples from an external store or *Take* apples from the other agent. A full description is provided in the supplementary information.

A selfish rational agent would never press the button in any of these games. This prediction was borne out by our A3C agent baseline (Figure 2). On the other hand, advantageous-inequity-averse

agents pressed their buttons significantly more often in the Give apples and Dictate apples games. They pressed the button even in the Dictate apples game when doing so could only reduce their own (extrinsic) payoff. Disadvantageous-inequity-averse agents pressed their button in the Take apples game to reduce the rewards obtained by the player with the larger initial endowment despite there being no extrinsic benefit to doing this.

## A.7 Theoretical arguments for the success of inequity aversion

We provide theoretical arguments for inequity aversion as an improvement to temporal credit assignment, extending the work of (Fehr and Schmidt 1999) beyond simple market games. In an intertemporal social dilemma, defection dominates cooperation in the short term. To leading order, the short-term Schelling diagram for an intertemporal social dilemma looks like Figure 3A, since by definition defection must dominate cooperation. Here and in the sequel we work in the limit of large number of players $N$. Mathematically, we denote defector payoff by $D$, cooperator payoff by $C$ and average payoff across the population by $\bar{R}$, writing:

$$C = c, \quad D = d, \quad \bar{R} = -\frac{d-c}{N}x + d, \quad \text{with } d > c. \tag{5}$$

First consider the effect of advantageous inequity aversion (AIA) on the short-term payoffs. Clearly the cooperator line is unchanged, since it is dominated. Hence the cooperator and defector lines become:

$$\tilde{C} = c, \quad \tilde{D} = D - \alpha(D - \bar{R}) = d - \alpha\frac{(d-c)}{N}x, \quad \text{with } \alpha > 0. \tag{6}$$

The transformed short-term payoffs are shown in Figure 3B. Since the $C$ curve dominates $\tilde{D}$ in some region, cooperative behavior can be self-sustaining in the short-term. Thus AIA improves temporal credit assignment. AIA can resolve the social dilemma when the earliest learned behavior generates *multiple* cooperators. This is the case for the Cleanup game but not the Harvest game, explaining the results.

The primary effect of disadvantageous inequity aversion (DIA) is to lower the payoff to a cooperator. However, it also motivates the cooperator to use the fining tool to reduce $\bar{R}$. There are several simple reasons why defectors might end up being especially targeted. Firstly, the behavior that avoids the policing agent may be cooperative (as in the Harvest game). Secondly, policing agents are motivated to avoid tagging other policers, because of the danger of retaliation.

Assuming that defectors are especially targeted, the cooperator and defector lines become:

$$\tilde{C} = C - \beta_C(\bar{R} - C) = c + \beta_C\left(\frac{d-c}{N}x - (d-c)\right), \tag{7}$$

$$\tilde{D} = D - \beta_D(\bar{R} - C) = d + \beta_D\left(\frac{d-c}{N}x - (d-c)\right), \tag{8}$$

with $\beta_D > \beta_C > 0$. The transformed short-term payoffs are shown in Figure 3C. Here the Nash equilibrium has moved to a positive number of cooperators. Hence DIA has improved temporal credit assignment. Of course, this argument requires the policing effect to emerge in the first place. This is possible when the earliest learned behavior is defection (Harvest), but not when it is cooperation (Cleanup), explaining the results.

Figure 3: Inequity aversion alters the effective payoffs from cooperation and defection in the short-term, in such a way that cooperative behavior is rationally learnable. Hence, helps to solve the intertemporal social dilemma.