[Reviews · NeurIPS 2018]

Reviewer 1



Summary The authors consider the addition of inequity aversion to learning agents in social dilemmas. They show that inequity aversion as a heuristic helps systems converge to cooperation. They show this is true even when only a few agents are inequity averse. Evaluation There is a lot to like about this paper. There are 2 main contributions: 1) Rather than considering end-to-end learning the authors consider simple heuristics that can be readily applied to solve a wide class of problems. 2) I also like the eligibility trace formulation of inequity aversion for Markov games, this opens, I think a wide way of experimenting with utility functions that include terms beyond one’s own reward in RL. I recommend acceptance of this paper. I have a few comments about the exposition for the authors that I think could improve the overall clarity of the paper. First, I think it would be good to state in the introduction what the point of the research program is. Is it: a) To design agents for environments where we control all agents, e.g. robot soccer (in which case, why not just optimize joint rewards?) b) To design agents for environments where we control some but not all agents? (In this case, isn’t advantageous inequity aversion a bad idea since it makes one very exploitable?) c) To study how certain heuristics can evolve in a population? (In this case, it would be good to see the payoffs of each type of agent in games with mixed populations since that’s what would be favored by evolution) d) Something else? I think nailing down more clearly in the introduction what the goal is and why people should care about this will make the paper even higher impact.

Reviewer 2



SUMMARY: the paper argues that incorporating inequality aversion (dislike to be advantageous or dislike to be disadvantageous) into the Markov games improves the long-term outcome for social groups STRENGTHS: -- The aversion concept is theoretically interesting and the authors show how it can be incorporated into Markov games -- The paper is well-written. It WEAKNESSES: -- There are no theoretical results. -- Experimental results do not convincingly show that inequality aversion will generally improve the outcome beyond selected games in selected settings. The results seem cherry-picking to me. -- When there are multiple players, the space of strategies become exponentially large. It is not clear that such a big space has been explored systematically yet. ----------------------------------- After the author's rebuttal ----------------------------------- I'm sorry for a delayed response. I agree with R1 that the heuristics in this paper can be applied to a wide range of problems. I adjusted my score accordingly. I read the author's rebuttal, and really like the new introduction, as the motivation of this paper is much clearer. I am a bit confused however why it makes sense to model the Schelling diagram as a linear function of the number of other cooperators.

Reviewer 3



This paper propose a model for the behavior of agents facing inter temporal social dilemmas. They found that inequity aversion resolve the intertemporal social dilemmas, i.e., agents in social dilemmas are able to find cooperative solutions with inequity aversion. This paper study an interesting and challenging problem in the field, and improves the prior works in different aspects as explained in the following. To model agents behavior it uses the multi-agent reinforcement learning in partially observable games presented in [40,41], and to learn the policies it uses the deep neural network introduced in A3C [42], but incorporate the subjective reward besides the regular rewards. This subjective reward is defined by their inequity aversion model, which is the generalization of the inequity aversion model [10] in static matrix games to the sequential multi-agents games.
They also formally defined the “sequential social delimma”, extending the definition of [17] which is for 2-player games and based on payoff matrix, but here it is defined based on Schelling diagram. Finally, in the experiments they show that inequity aversion improves agents cooperation in some example multi-person games with social dilemma (according to their conditions in sec 2.3).